# GEOMETRIC AUGMENTATION FOR ROBUST NEURAL NETWORK CLASSIFIERS

## ABSTRACT

We introduce a novel geometric perspective and unsupervised model augmentation framework for transforming traditional deep (convolutional) neural networks into adversarially robust classifiers. Class-conditional probability densities based on Bayesian nonparametric mixtures of factor analyzers (BNP-MFA) over the input space are used to design soft decision labels for feature to label isometry. Class-conditional distributions over features are also learned using BNP-MFA to develop plug-in maximum a posterior (MAP) classifiers to replace the traditional multinomial logistic softmax classification layers. This novel unsupervised augmented framework, which we call geometrically robust networks (GRN), is applied to CIFAR-10, CIFAR-100, and to Radio-ML (a time series dataset for radio modulation recognition). We demonstrate the robustness of GRN models to adversarial attacks from fast gradient sign method, Carlini-Wagner, and projected gradient descent.

## 1 INTRODUCTION

DeepConvNets are already prevalent in speech, vision, self-driving cars, biometrics, and robotics. However, they possess discontinuities that are easy targets for attacks as evidenced in dozens of papers (see (Goodfellow et al., 2015; Nguyen et al., 2015; Papernot et al.) and references therein). Adversarial images can be made to be robust to translation, scale, and rotation (Athalye et al., 2017). Adversarial attacks have also been applied to deep reinforcement learning (Huang et al., 2017; Kos & Song, 2017) and speech recognition (Carlini & Wagner, 2018a). In this work we will also consider attacks on automatic modulation recognition using deep convolutional networks (O'Shea et al., 2016). Previous work in creating adversarially robust deep neural network classifiers includes robust optimization with saddle point formulations (Madry et al., 2018), adversarial training (see e.g., (Kurakin et al., 2017)), ensemble adversarial training (Tramer et al., 2018), defensive distillation (Papernot et al., 2016), and use of detector-reformer networks (Meng & Chen, 2017). Defensive distillation has been found to be an insufficient defense (Carlini & Wagner, 2016; 2017) and MagNet of (Meng & Chen, 2017) was also shown to be defeatable in (Carlini & Wagner, 2018b). A summary of the attacks and defenses from the NIPS 2017 competition on adversarial attack and defense can be found in (Kurakin et al., 2018).

In this paper we propose a statistical geometric model augmentation approach to designing robust neural networks. We argue that signal representations involving projections onto lower-dimensional subspaces lower mean square error distortion. We implement a statistical union of subspaces learned using a mixture of factor analyzers to create the auxiliary signal space structural information that neural networks can use to improve robustness. We use the geometry of the input space to create unsupervised soft probabilistic decision labels to replace traditional hard one-hot encoded label vectors. We also use the geometry of the feature space (after soft-decision supervised training) to create accurate class-conditional probability density estimates for MAP classifiers (to replace neural network classification layers). We call this unsupervised geometric augmentation framework geometrically robust networks (GRN). The main contributions of this paper are:

1. Geometric analysis of problems with current neural networks.

2. A novel soft decision label coding framework using unsupervised statistical-geometric union of subspace learning.

   3. Maximum a posteriori classification framework based on class-conditional feature vector density estimation.

The rest of this paper is organized as follows. In Section 2 we analyze neural networks from a geometric vantage point and recommend solution pathways for overcoming adversarial brittleness. In Section 3 we describe the full details of the proposed geometrically robust network design framework. We give experimental results on two datasets and three attacks in Section 4 and conclude in Section 5.

## 2   ANALYSIS OF NEURAL NETWORKS FROM GEOMETRIC VIEWPOINT

A deep (convolutional) neural network is a nested nonlinear function approximator that we can write as

$$\mathbf{g}_\Theta(\mathbf{x}) = \underbrace{\mathbf{h}_{\theta_C^c}(\mathbf{h}_{\theta_{C-1}^c}(\cdots \mathbf{h}_{\theta_1^c}(}_{\text{classification}} \underbrace{\mathbf{h}_{\theta_F^f}(\mathbf{h}_{\theta_{F-1}^f}(\cdots \mathbf{h}_{\theta_1^f}(\mathbf{x}))))))}_{\text{feature extraction}}) \tag{1}$$

where in our notation $\theta^c$ denotes parameters (weights and biases) associated with the classification stages $c = 1, 2, ..., C$, $\theta^f$ denotes parameters associated with the feature extraction stages $f = 1, 2, ..., F$, and the parameters are nested unions as $\theta_l \supset \theta_{l-1}$ culminating in $\Theta$. In principle an ideal function $\mathbf{g}^*$ which could be resilient to bounded adversarial noise is guaranteed to exist by the universality theorem for neural networks (Cybenko, 1992), so this drives an investigation into what is making current architectures brittle.

### 2.1   NEED FOR SOFT DECISION LABELS

The cross entropy loss objective function typically used to train function approximator $\mathbf{g}_\Theta \in \mathbb{R}^K$ ($K$ classes) in (1) on $n$ samples $\{\mathbf{x}_i\}_{i=1}^n$ with label vectors $\mathbf{y}(\mathbf{x}_i)$ has the form:

$$\underset{\Theta}{\text{argmin}} -\frac{1}{n} \sum_{i \in \{1,...,n\}} \sum_{k \in \{1,...,K\}} y_k(\mathbf{x}_i) \log g_{k,\Theta}(\mathbf{x}_i) \tag{2}$$

For hard decision labels, $\mathbf{y}(\mathbf{x}_i)$ is an indicator vector (i.e. one-hot encoding), and (2) collapses to $\text{argmax}_\Theta \, n^{-1} \sum_{i \in \{1,...,n\}} \log g_{t(x_i),\Theta}(\mathbf{x}_i)$ where $t(\mathbf{x}_i)$ is the element in indicator vector $\mathbf{y}(\mathbf{x}_i)$ that is equal to one for the $i^{th}$ sample $\mathbf{x}_i$. Following the analysis in (Papernot et al., 2016), this means the stochastic gradient descent training algorithm minimizing (2) will inherently constrain the weights $\theta^c$ in the final classification layer(s) of (1) to try to output zeros for elements not corresponding to the correct class. This artificially contrains the network to be overconfident and introduces brittleness.

There is also a geometric argument against one-hot encoding. The mapping from feature space to label space is a surjective map since the mapping to hard decisions reduces the set size of the codomain to be equal to the number of classes, and the number of unique features will generally be larger than the number of unique classes. This prevents the formation of an injective map from feature space to estimated label space. If we can enlarge the set of labels to be infinite (i.e. soft decisions), then we allow room for an injective map to be learned from feature to estimated label space. The resulting bijection (albeit a nonlinear bijection) then opens the door for distance preserving maps (isometries) which can guarantee that small distances between points in feature space remain small distances in the label space. This kind of isometry is exactly what we need to be able to have adversarially robust networks.

### 2.2   NEED FOR GEOMETRIC MODELS

Letting $\|\cdot\|_2$ denote $L_2$-norm, $\mathbf{P}_{(\cdot)}$ a projection matrix, $\mathbf{P}_{(\cdot)}^\perp$ the orthogonal complement, $\mathbf{x}$ a natural input, and $\mathbf{x}_{adv}$ the adversarially perturbed input, we know that $\|\mathbf{x} - \mathbf{x}_{adv}\|_2 \geq \|\mathbf{P}\mathbf{x} - \mathbf{P}\mathbf{x}_{adv}\|_2$ since $\|\mathbf{x}\|_2^2 = \|\mathbf{P}\mathbf{x}\|_2^2 + \|\mathbf{P}^\perp\mathbf{x}\|_2^2$ by Pythagorean theorem. If the data $\mathbf{x}$ is well approximated by an information-preserving projection into another subspace, then we can reduce distortion of the adversarial input in the projected latent space. If a network can be made to exploit knowledge of latent spaces with distortion-reducing representations of the data, then the overall classification performance would be less sensitive to adversarial perturbations. A density estimate built upon this geometrical structure would then implicitly capture projected data representations and ultimately minimize the

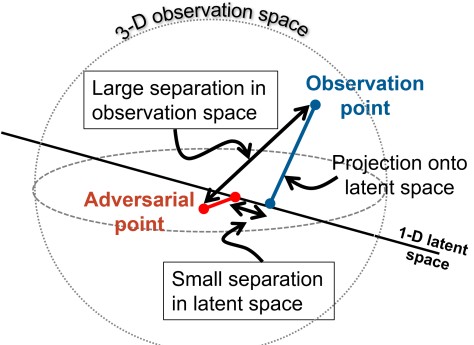

(a) Conceptual illustration of how large deviations in the ambient observation space can translate into small deviations in a lower dimensional latent space that captures most of the signal information.

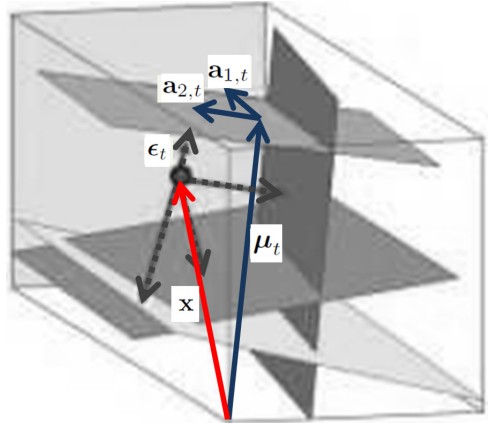

(b) Illustration of a union of subspaces depicted by 2-d overlapping planes.

Figure 1: Geometric modeling of signals as points that lie close to one or more linear subspaces. The signal data (either input observations to a neural network or feature vectors learned from the network) are modeled as small error displacements from a low dimensional linear subspace. These union of subspaces can be learned statistically using a mixture of factor analyzers. Since the number of subspaces and dimensionality of each subspace are unknown a priori, we must use a Bayesian nonparametric model.

label space deviation. The vast majority of current deep neural network models make no use of geometrical-statistical models of the data and are solely supervised learning on labeled inputs. We must use unsupervised learning to learn the latent manifold or union of subspaces topology to assist the supervised learning piece. The latent structure of the data can be captured in both the input space and feature space as we will do in this study.

Here we briefly introduce the union of subspaces (UoS) model for modeling inputs and features. To illustrate this, we take a vectorized signal segment $\mathbf{x}$ as shown in Figure 1(b) as a $D$-dimensional point living close to a union of $T$ linear subspaces

$$\mathbf{x} \in \cup_{t=1}^{T}(\mathcal{S}_t + \boldsymbol{\epsilon}_t) \in \mathbb{R}^D \tag{3}$$

where

$$\mathcal{S}_t = \{\mathbf{A}_t\mathbf{w} + \boldsymbol{\mu}_t : \mathbf{w} \in \mathbb{R}^{d_t}\} \quad . \tag{4}$$

The matrix $\mathbf{A}_t = [\mathbf{a}_{1,t}, \mathbf{a}_{2,t}, ..., \mathbf{a}_{d_t,t}]$ is the matrix of basis vectors centered at $\boldsymbol{\mu}_t$ for subspace index $t$, $\mathbf{w}$ is the coordinate of $\mathbf{x}$ at $t$, and $\boldsymbol{\epsilon}_t \sim \mathcal{N}(0, \sigma_\epsilon^2\mathbf{I})$ is the modeling error. The subspace coordinates $\mathbf{w}_i$ and the closest subspace index $t(i)$ are the latent variables for observation $\mathbf{x}_i$. In Figure 1(b), we show the signal vector $\mathbf{x}$, subspace offset vector $\boldsymbol{\mu}_t$, local basis vectors $\mathbf{a}_{jt}$, and modeling error $\boldsymbol{\epsilon}_t$. The locus of all potential signal vectors of interest is assumed to lie on or near one of the local subspaces. Since the observation is assumed to lie close to one of the $T$ subspaces we can therefore write the $i^{th}$ observation as

$$\mathbf{x}_i = \mathbf{A}_{t(i)}\mathbf{w}_i + \boldsymbol{\mu}_{t(i)} + \boldsymbol{\epsilon}_{t(i)} \tag{5}$$

## 3 GEOMETRICALLY ROBUST NETWORKS

In this section we provide the complete framework for inserting our statistical-geometric viewpoint from Section 2 into a robust design approach that we will call geometrically robust networks (GRN). We propose to use unsupervised learning on the input space for label encoding and unsupervised learning on the feature space for density estimation in a MAP classifier. In this study we target the classification layers in $\mathbf{g}_\Theta$ of (1) as the key layers for improvement assuming the supervised feature learning has adequately reached the information bottleneck limit Shwartz-Ziv & Tishby (2017). We will improve upon the classification layers in two fundamental ways:

1. Use soft decision labels to train the neural network.

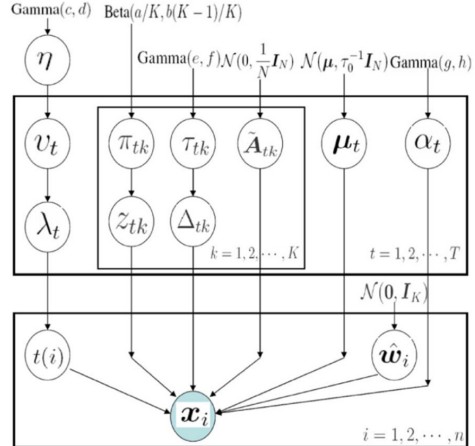
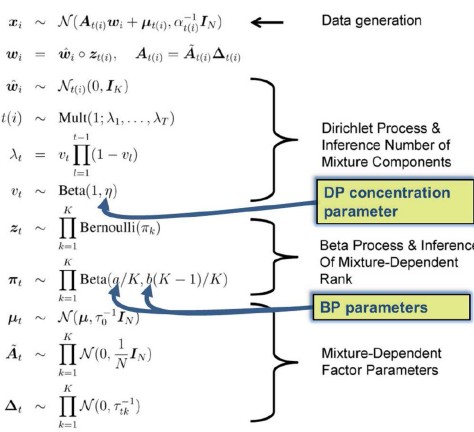

(a) Graphical model of BNP-MFA (Chen et al., 2010).

(b) Hierarchical roll out of conjugate exponential BNP-MFA model.

Figure 2: The Bayesian nonparametric mixture of factor analyzers (BNP-MFA) from (Chen et al., 2010) which is our building block for estimating the union of subspaces. The tunable hyperparameters are the Dirichlet process (DP) concentration parameter which influences the number of mixtures/subspaces and the Beta process (BP) parameters which influence the dimensionality of each subspace.

2. Replace the classification layers $\mathbf{h}_{\theta_C^c}, \mathbf{h}_{\theta_{C-1}^c}, ..., \mathbf{h}_{\theta_1^c}$ which generally implement a softmax multinomial logistic regression with a Bayesian maximum a posteriori (MAP) classifier using plug-in class-conditional density estimates.

In Subsection 3.1 we summarize the Bayesian nonparametric mixture of factor analyzers (BNP-MFA) model for union of subspace learning. In Subsection 3.2 we describe label encoding and MAP classifition steps which directly follow from learning the BNP-MFA.

## 3.1 BAYESIAN NONPARAMETRIC MIXTURE OF FACTOR ANALYZERS

To estimate the geometric model described above in Section 2.2 we use the Bayesian nonparametric formulation of the mixture of factor analyzers (BNP-MFA introduced in (Chen et al., 2010)) which has several advantages for estimating our required statistical union of subspaces:

1. *Accuracy*: Mixture of factor analyzer models empirically show higher test log-likelihoods (model evidence) than Boltzmann machine based models (Tang et al., 2012).

2. *Speed*: Since the BNP-MFA is a conjugate-exponential model it can be learned in online variational Bayes form with stochastic variational inference (Hoffman et al., 2013) giving it orders of magnitude speed improvements compared to Markov chain Monte Carlo methods.

3. *Scales with data*: Since the model is Bayesian nonparametric, there is no model overfitting and no need for regularization.

4. *Hyperparameter Insensitive* Only two hyperparameters need to be set are they are very insensitive to overall performance.

Under the BNP-MFA framework we infer the number of subspaces and subspace rank from the data using Bayesian nonparametrics. A Dirichlet process mixture model is used to model the clusters, while a Beta process is used to estimate the local subspace in each cluster. The conjugate-exponential directed graphical model shown in Figure 2(a) and hierarchical roll out in Figure 2(b) is taken from (Chen et al., 2010). Here, the $\{\mathbf{x}_i\}_{i=1}^n$ are vector-valued observations in $\mathbb{R}^N$ with component weights given by the vectors $\{\hat{\mathbf{w}}_i\}_{i=1}^n$ in $\mathbb{R}^K$, $\{\tilde{\mathbf{A}}_t, \Delta_t, \tau_t, \mathbf{z}_t, \pi_t, \mu_t, \alpha_t, v_t\}_{t=1}^T$ are various global parameters for each of the $T$ mixture components, $\eta \in \mathbb{R}$ is a global parameter for the Dirichlet process, $\boldsymbol{\mu} = \frac{1}{n}\sum_{i=1}^n x_i$ is the (fixed) sample mean of the data and $a$-$h$ and $\tau_0$ are fixed constants.

For each $t$, the global variables have dimensions $\tilde{\mathbf{A}}_t \in \mathbb{R}^{N \times K}$, $\Delta_t \in \mathbb{R}^{K \times K}$, $\tau_t \in \mathbb{R}^K$, $\mathbf{z}_t \in \mathbb{R}^K$, $\pi_t \in \mathbb{R}^K$, $\mu_t \in \mathbb{R}^N$, $\alpha_t \in \mathbb{R}$ and $v_t \in \mathbb{R}^T$. More details on this model and the motivation for its construction can be found in (Chen et al., 2010) After the BNP-MFA model finishes training, we have the all the parameters (centroids, subspace spanning vectors, and cluster weights) that we need to estimate the class conditional probability density function (6) which we will use for both MAP classification and soft-decision label encoding as we show in section 3.

## 3.2    Soft Decision Labels and MAP Classifiers from Geometric Model

The idea of soft decision labels was used in defensive distillation (Papernot et al., 2016). Papernot et. al. used the first pass through their target neural network $\mathbf{g}(\mathbf{x})$ with annealed softmax to learn to the soft decision labels $\mathbf{y} = \mathbf{g}(\mathbf{x})$. They then used those learned labels $\mathbf{y}$ in the second pass through the same network but with different softmax thermal parameters to create the distilled network. As pointed out in (Papernot et al., 2016), the distilled network $\mathbf{g}^d(\cdot)$ will converge toward the originally network $\mathbf{g}(\cdot)$ under a cross entropy loss given enough training data. Thus, the distilled network can still possess some of the brittle nature of the original network trained with hard decision labels. This vulnerability was revealed to be the case in (Carlini & Wagner, 2016; 2017).

We deviate from Papernot's defensive distillation approach here by using class conditional density estimate $\hat{p}_\Psi(\mathbf{x}|k)$ on the class-partitioned input data with $K$ total classes to create our labels. Here, we use the fact that the BNP-MFA is a demarginalization of a Gaussian mixture model (GMM) to form density estimates. The term demarginalization from Robert & Casella (2005) is taken here to mean the formation of a latent variable probability density which is the integrand under a marginalization integral. We learn the BNP-MFA model with parameters $\Psi$ from the original class-partitioned signals/images as input and then estimate the class-conditional pdf over the input space as

$$\hat{p}_\Psi(\mathbf{x}|k) = \sum_{t=1}^{T} \lambda_{kt} \int \mathcal{N}\left(\mathbf{x}; \tilde{A}_{kt}(\boldsymbol{\Delta}_{kt}\mathrm{diag}(z_{ki}))\hat{\mathbf{w}}_k + \boldsymbol{\mu}_{kt}, \alpha_{kt}^{-1}\mathbf{I}_N\right) \mathcal{N}\left(\hat{\mathbf{w}}_k; \boldsymbol{\xi}_{kt}, \boldsymbol{\Lambda}_{kt}\right) d\hat{\mathbf{w}}_k$$

$$= \sum_{t=1}^{T} \lambda_{kt} \mathcal{N}\left(\mathbf{x}; \boldsymbol{\chi}_{kt}, \boldsymbol{\Omega}_{kt}\right)$$

$$\boldsymbol{\chi}_{kt} = \boldsymbol{\mu}_{kt} + \tilde{\mathbf{A}}_{kt}\boldsymbol{\Delta}_{kt}\mathrm{diag}(z_{ki})\boldsymbol{\xi}_{kt}$$

$$\boldsymbol{\Omega}_{kt} = \tilde{\mathbf{A}}_{kt}\boldsymbol{\Delta}_{kt}\mathrm{diag}(z_{ki})\boldsymbol{\Lambda}_{kt}\mathrm{diag}(z_{ki})\boldsymbol{\Delta}_{kt}\tilde{\mathbf{A}}_{kt}^T + \alpha_{kt}^{-1}\mathbf{I}_N \tag{6}$$

Then, we assign our label vector as the posterior

$$y_{ki} = \beta_{k_i}\hat{p}_\Psi(\mathbf{x}_i|k)p(k) \quad \forall i = 1, 2, ..., n \quad \text{and} \quad \forall k = 1, 2, ..., K \tag{7}$$

where $p(k)$ is the class prior. The term $\beta_{k_i}$ is a combined correction factor and normalization factor to scale the correct class label higher than incorrect classes for the cases where $y_i[k] \geq y_i[k^*]$ and where $k^*$ is the correct class index. The $\beta_{k_i}$ term also enforces that $\sum_k y_{ki} = 1, \forall i$. Soft decision label encoding based on class conditional likelihood has the advantage that it is independent of any deep architecture.

Once we learn the labels $\{\mathbf{y}_i\}_{i=1}^n$ from (7), we apply those labels to learn the neural network $\mathbf{g}_\Theta$ from $\mathbf{g}_\Theta(\mathbf{x}_i) = \mathbf{y}_i, \forall i = 1, ..., n$ using traditional backprogation with SGD training on a cross entropy loss function. After the model converges, we extract features $\mathbf{z}_i = \mathbf{h}_{\Theta_f}(\mathbf{x}_i), \forall i = 1, ..., n$ and learn a new BNP-MFA with model parameters $\Phi$ over the feature space. To learn the feature space class-conditional pdfs we simply swap out $\mathbf{x}$ with $\mathbf{z}$ and the $\Psi$ with $\Phi$ in (6) to obtain the approximate class-conditional likelihood functions over the feature vectors. The approximate posterior pdf is then simply $\hat{p}_\Phi(k|\mathbf{z}) \propto \hat{p}_\Phi(\mathbf{z}|k)p(k)$. For the datasets we benchmark over the prior $p(k) = 1/K$ is uniform, and the MAP classifier reduces to maximum likelihood (ML) classification. However, in the real world class priors are almost never uniform and MAP classification gives a significant boost not only over multinomial logistic regression but ML classification as well. We summarize the training and testing stage procedures of GRN in Algorithm 1.

In Figure 3 we show plots of the negative squared Mahalanobis distance for each cluster of the class conditional input space pdf in (6) for a single image confuser sample from a Carlini-Wagner attack compared to the original unperturbed image. We see that the adversarial attack has almost no

---

**Algorithm 1** Geometrically Robust Networks (GRN) Augmentation Framework

---

1: **procedure** TRAINING PHASE SUMMARY
2:      Given $\{\mathbf{x}_i, k_i\}_{i=1}^n$ learn $\hat{p}_\Psi(\mathbf{x}|k)$ using BNP-MFA
3:      Label encode: $y_{ki} = \beta_{k_i}\hat{p}_\Psi(\mathbf{x}|k)p(k) \quad \forall i = 1,...,n$
4:      Learn base model $\{\mathbf{g}_\Theta(\mathbf{x}_i) = \mathbf{y}_i\}_{i=1}^n$ using SGD backprop on cross entropy loss to select feature extraction layer $\mathbf{h}_{\Theta_f}(\mathbf{x}_i)$ in (1)
5:      Extract features: $\{\mathbf{z}_i = \mathbf{h}_{\Theta_f}(\mathbf{x}_i)\}_{i=1}^n$
6:      Given $\{\mathbf{z}_i, k_i\}_{i=1}^n$ learn $\hat{p}_\Phi(\mathbf{z}|k)$ using BNP-MFA
7: **end procedure**
8: **procedure** TESTING PHASE SUMMARY
9:      Given test inputs $\{\mathbf{x}_i\}_{i=1}^{n_{test}}$, nonlinear function $\mathbf{h}_{\Theta_f}(\cdot)$, class prior $p(k)$, and pdf estimate $\hat{p}_\Phi(\cdot|k)$, estimate class label as $\hat{k}_i = \mathrm{argmax}_k\,\hat{p}_\Phi(\mathbf{h}_{\Theta_f}(\mathbf{x}_i)|k)p(k)$
10: **end procedure**

---

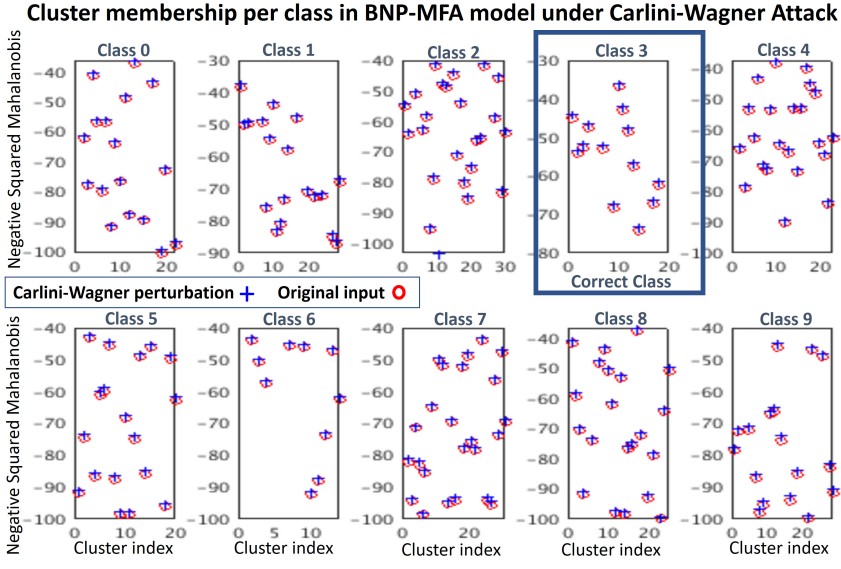

Figure 3: Matrix of ten plots (one for each of the ten CIFAR-10 classes) showing the negative squared Mahalanobis distance for each cluster of the class conditional input space pdf in (6) for a single image confuser sample from a Carlini-Wagner attack compared to the original unperturbed image. We see that the adversarial attack has almost no influence on the pdf components.

influence on the pdf in (6) and, therefore, practically no variation on the corresponding label in (7). This demonstrates the concept depicted in Figure 1(a) of how the projected data points have very little deviation in the latent space.

## 4   RESULTS

For our base neural network from which we build the GRN in step 4 of Algorithm 1 for CIFAR-10 and CIFAR-100 we use Springenberger's "All convolutional network" (Springenberg et al., 2015) which uses only convolutional layers for entire stack. Our black box network from which we craft adversarial samples for CIFAR-10 and CIFAR-100 is: [3x3 conv 32 LeakyReLU(0.2), 3x3 conv 32 LeakyReLU(0.2), 2x2 MaxPool Dropout(0.2), 3x3 conv 64 LeakyReLU(0.2), 3x3 conv 64 LeakyReLU(0.2), 2x2 MaxPool Dropout(0.3), 3x3 conv 128 LeakyReLU(0.2), 3x3 conv 128 LeakyReLU(0.2), 2x2 MaxPool Dropout(0.4), Dense 512 ReLU Dropout(0.5), Dense 10 Softmax].

The Radio-ML dataset `https://radioml.com/datasets` is a relatively new time series dataset for benchmarking radio modulation recognition tasks. It has 11 modulation schemes (3

Table 1: Probability of Correct Classification (all attacks black box)

| Dataset | Model | No Attack | FGSM | Carlini-Wagner | PGD |
|---------|-------|-----------|------|----------------|-----|
| CIFAR-10 | Geometrically Robust Network | 0.84 | **0.81** | **0.72** | **0.75** |
|  | All-Conv-Net | 0.9 | 0.17 | 0.12 | 0.13 |
| CIFAR-100 | Geometrically Robust Network | 0.51 | **0.49** | TBD | TBD |
|  | All-Conv-Net | 0.52 | 0.37 | TBD | TBD |

analog, 8 digital) undergoing sample rate offset, center frequency offset, frequency flat fading, and AWGN. We measure probability of correct classification as a function of signal-to-noise ratio (SNR) for that dataset. For the Radio-ML dataset our base neural network from which we build the GRN in step 4 of Algorithm 1 is given at the top of Figure 4(b). The black box attack network for Radio-ML is a version of LeNet-5 CNN used in (O'Shea et al., 2016) and is shown at the bottom of Figure 4(b).

For both datasets the data was scaled to lie between zero and one with respect to the adversarial parameter settings. Using cleverhans (Nicolas Papernot, 2017), we craft adversarial samples from fast gradient sign method (FGSM), Carlini Wagner (CW) (Carlini & Wagner, 2017), and projected gradient descent (PGD) (Madry et al., 2018) on the CIFAR-10 and CIFAR-100 dataset. With FGSM we use eps=.005. With CW method we use initial tradeoff-constant = 10, batch size = 200, 10 binary search steps, and 100 max iterations. With PGD we use eps = 1, number of iterations = 7, and a step size for each attack iteration = 2. These attack parameter settings were more than enough to confuse the base classifier while keeping the mean structural similarity index (Wang et al., 2004) relatively constant between natural and adversarial images. For the Radio-ML dataset we only experiment with FGSM (eps=.03). For the CIFAR-100 dataset we used the following data augmentation parameters: (1) 10 degree random rotations, (2) zoom range form .8 to 1.2, (3) width shift range percentatage = 0.2, (4) height shift range percentage = 0.2, and (5) random horizontal flips.

As shown in Table 1 the proposed GRN model for CIFAR-10 performed remarkably well in the face of all three attacks suffering only about 10-20 percent accuracy compared to base network with no attack (natural test samples only). (At the time of this submission we are running the CW and PGD attacks on CIFAR-100 and plan to include results on next iteration of paper.) The CIFAR-100 results under no attack did not match the reported results in (Springenberg et al., 2015) likely because we did not use as extensive data augmentation. In Figure 4(a) we plot the accuracy versus SNR for the four cases using the Radio-ML dataset: 1) base model no attack, 2) base model FGSM attack, 3) GRN model no attack, and 4) GRN model FGSM attack. Again, we see that the GRN is relatively unaffected by the adversarial attack. We also experimented with hyperparameter settings for the Dirichlet process concentration parameter $\eta$ and the ratio of the Beta process parameters $\frac{a}{b}$. We observed that up to one order of magnitude in change in $\eta$ and $\frac{a}{b}$ there was demonstrable change in the ultimate classification performance.

## 5 CONCLUSION AND FUTURE WORK

We have demonstrated that geometrical statistically augmented neural network models can achieve state-of-the-art robustness on CIFAR-10 under three different adversarial attack methods. We hope that this work will be the start of further investigation into the idea of using geometrically centered unsupervised learning methods to assist in making deep learning models robust, not only to adversarial noise but to all types of noise. There is more work that could be done to understand the best way to engineer soft decision labels given auxiliary data models. We need to also understand if the training algorithms themselves can be directly manipulated to incorporate outside structural data models.

A main selling point of Bayesian nonparametrics has been that the complexity of the model can grow as more data is observed. However, the current training algorithm for the BNP-MFA model is Gibbs sampling, which fails to scale to massive data sets. Stochastic variational inference (Hoffman et al., 2013) has been introduced as one such way to perform variational inference for massive or streaming data sets. We are currently working to cast the BNP-MFA into a stochastic variational framework so that the GRN model can be extended to very large (or even streaming) datasets.

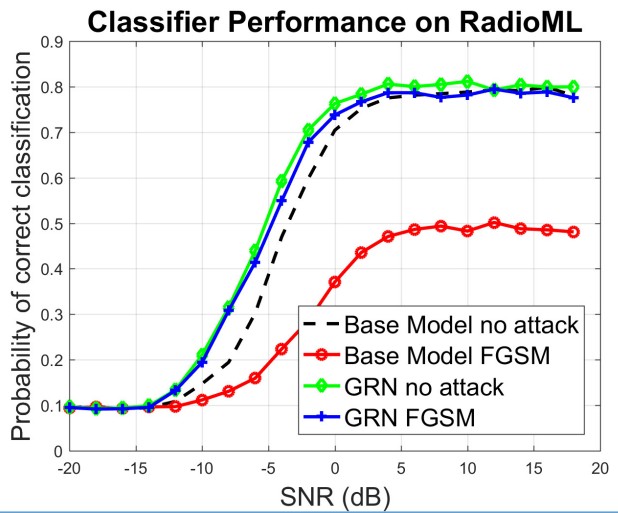

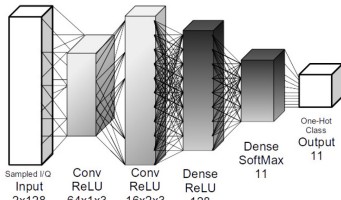

Base Defense Network

| 1x3 conv 256 ReLU dropout(.5) |
| 2x3 conv 80 ReLU dropout(.5) |
| Dense 256 ReLU Dropout(0.5) |
| Dense 64 ReLU Dropout(0.5) |
| Dense 32 ReLU Dropout(0.5) |
| Dense 11 ReLU Softmax |

Black Box Network (LeNet-5)

(a) Probability of correction classification vs SNR for the base model and proposed GRN model.

(b) (Top) The base defense network for Radio-ML from which we build our GRN in Algorithm 1. (Bottom) The black box network used to craft the FGSM attack for Radio-ML (modified LeNet-5 from (O'Shea et al., 2016).)

Figure 4: Network specification and performance results for proposed geometrically robust networks applied to the Radio-ML dataset (modulation recognition over 11 modulation formats).

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
