# OpenReview forum: "GEOMETRIC AUGMENTATION FOR ROBUST NEURAL NETWORK CLASSIFIERS"
_ICLR.cc/2019/Conference_

### Official Review · AnonReviewer3 · 2018-10-31
**This work lacks any convincing experimental result to support the claims**

**Rating:** 3
**Confidence:** 5

**Review:**

This work proposes a defence based on class-conditional feature distributions to turn deep neural networks into robust classifiers.

At present this work lacks even the most rudimentary evidence to support the claims of robustness, and I hence refrain from providing a full review. In brief, model robustness is only tested against adversarials crafted from a standard convolutional neural network (i.e. in a transfer setting, which is vastly different from what the abstract suggests). Unsurprisingly, the vanilla CNN is less robust than the density-based architecture introduced here, but that can be simply be explained by how close the substitute model and the vanilla CNN are. No direct attacks - neither gradient-based, score-based or decision-based attacks - have been used to evaluate robustness. Please check [1] for how a thorough robustness evaluation should be performed.

[1] Schott et al. “Towards the first adversarially robust neural network model on MNIST”.

---

### Official Review · AnonReviewer2 · 2018-11-02
**Interesting work but more comprehensive evaluations needed**

**Rating:** 4
**Confidence:** 4

**Review:**

This paper proposes geometrically robust networks (GRN), which applies geometric perspective and unsupervised model augmentation to transform traditional deep neural networks into adversarial robust classifiers. Promising experimental results against several adversarial attacks are presented as well.

The BNP-MFA are applied twice in the framework: one for getting the soft labels, and the other for getting the predictions through MAP estimation. There are existing works which are in the same line as the second part: deep kNN [1], and simple cache model [2] for example, where similarities to training examples are used to derive the test prediction and substantial increase of the robustness against adversarial attacks considered in this work have also been shown.

These raise two questions:
(1) How much does the soft label encoding help increase the robustness?
(2) How does the proposed model compare with the deep kNN and the simple cache model, which are much simpler?

Some minor issues:
- The unsupervised learning for label encoding is performed on the input space, the image pixel for example. But it is known that they are not good features for image recognition.
- It is unclear which part of the network is considered as "feature extraction" part which is used for MAP estimation in the experiments.
- It would be nicer to have results with different architectures.


[1] N. Papernot and P. McDaniel. Deep k-nearest neighbors: towards confident, interpretable and robust deep learning. arXiv:1803.04765.
[2] E. Orhan. A simple cache model for image recognition. arXiv:1805.08709.

---

### Official Review · AnonReviewer4 · 2018-11-10
**The paper can be much improved by providing more evidence of the robustness to adversarial attack and advantages over other models.**

**Rating:** 4
**Confidence:** 3

**Review:**

The paper is working on a robust classifier that consists of two stages. The first stage performs unsupervised conditional kernel density estimates (KDE) of the covariate vectors, and the second stage is feature extractions and classification. I appreciate the authors' efforts to clarify the intuition, but more technical details and experiments can be provided to support their arguments. My questions and comments are below.

1. Page 2. "this means the stochastic gradient descent training algorithm minimizing..." Is the problem because of SGD or the structure of NN? I think the reason might be the latter, consider logistic regression, which can be seen as a single-layer NN, does not suffer such a problem.
2. I know the KDE part is from an existing paper, but more technical details can make the paper clearer and some statements are questionable. Specifically, what basis vectors are used for (3)? Is it really speedy and scalable (Page 4, Section 3.1) for BNP-MFA if using Gibbs sampling? Is it the reason why the experiments in Table 1 is incomplete?
3. For Eqn (7), how do you calculate \beta's to "scale the correct class label higher than incorrect classes for the cases...?"
4. Is the proposed model robust to all kinds of attacks, like gradient based noise, and outliers which locates far away from the corresponding cluster?
5. Can you provide some experiments to show the advantage over other approaches?[1]


I highly encourage the use of BNP KDE which has many advantages as stated in the paper. But the authors may have to solve the problem of scalability and show advantages over other approaches.

[1]Uličný, Matej, Jens Lundström, and Stefan Byttner. "Robustness of deep convolutional neural networks for image recognition." International Symposium on Intelligent Computing Systems. Springer, Cham, 2016.

---

### Meta-Review · Area_Chair1 · 2018-12-01
**More convincing experiments are needed**

**Confidence:** 5
**Recommendation:** Reject

**Metareview:**

All three reviewers feel that the paper needs to provide more convincing results to support their robustness claim, in addition to a number of other issues that need to be clarified/improved. The authors did not provide any response.